# Risk factors for COVID-19 mortality among telehealth patients in Bangladesh: A prospective cohort study

Ayesha Sania[1]ⓘ☼*, Ayesha S. Mahmud[2]ⓘ☼, Daniel M. Alschuler[1]ⓘ, Tamanna Urmi[3]ⓘ, Shayan Chowdhury[1,4], Seonjoo Lee[1]ⓘ, Shabnam Mostari[4], Forhad Zahid Shaikh[4], Kawsar Hosain Sojib[4,5]ⓘ, Tahmid Khan[6], Yiafee Khan[4]ⓘ, Anir Chowdhury[4], Shams el Arifeen[7]

**1** Department of Psychiatry, Columbia University Irving Medical Center, New York, New York, United States of America, **2** Department of Demography, University of California, Berkeley, Berkeley, California, United States of America, **3** Network Science Institute, Northeastern University, Boston, Massachusetts, United States of America, **4** Aspire to Innovate (a2i) ICT Division, Dhaka, Bangladesh, **5** Department of Economics, Jahangirnagar University, Dhaka, Bangladesh, **6** Department of Epidemiology, Columbia University Irving Medical Center, New York, New York, United States of America, **7** Maternal and Child Health Division, International Centre for Diarrhoeal Disease Research, Bangladesh, Dhaka, Bangladesh

☼ These authors contributed equally to this work.
* as4823@cumc.columbia.edu

**Data Availability Statement:** Data can be accessed by requesting A2i (the Aspire to Innovate program; https://a2i.gov.bd). Email to: tanvir.quader@a2i.gov.bd; shabnam.mostari@a2i.gov.bd Data set

## Abstract

### Background and objective

Estimating the contribution of risk factors of mortality due to COVID-19 is particularly important in settings with low vaccination coverage and limited public health and clinical resources. Very few studies of risk factors of COVID-19 mortality used high-quality data at an individual level from low- and middle-income countries (LMICs). We examined the contribution of demographic, socioeconomic and clinical risk factors of COVID-19 mortality in Bangladesh, a lower middle-income country in South Asia.

### Methods

We used data from 290,488 lab-confirmed COVID-19 patients who participated in a telehealth service in Bangladesh between May 2020 and June 2021, linked with COVID-19 death data from a national database to study the risk factors associated with mortality. Multivariable logistic regression models were used to estimate the association between risk factors and mortality. We used classification and regression trees to identify the risk factors that are the most important for clinical decision-making.

### Findings

This study is one of the largest prospective cohort studies of COVID-19 mortality in a LMIC, covering 36% of all lab-confirmed COVID-19 cases in the country during the study period. We found that being male, being very young or elderly, having low socioeconomic status, chronic kidney and liver disease, and being infected during the latter pandemic period were significantly associated with a higher risk of mortality from COVID-19. Males had 1.15 times

name: COVIDeatth_Bangladesh.csv
Variable list: sex; division; oc.any.death; oc.any. hospitalization; oc.any.hospitalization.or.death; any. q.Symptom.Fever; any.q.Symptom.Breath. Diff; any.q.Symptom.Headache; any.q.Symptom. Weakness; any.q.Symptom.Diarrhea; any.q. Symptom.Body.Ache; any.q.Home.Rules.Distance. Family; any.q.Home.Rules.Mask; any.q.Health. Improvement; any.q.Caregiver; any.q.MedHx. Diabetes; any.q.MedHx.Breath; any.q.MedHx.High. BP; any.q.MedHx.Other; any.q.MedHx.Kidney; any. q.MedHx.Liver; any.q.Travel.Abroad; any.q. Crowded.Place; any.q.Enough.Sleep; any.q.COVID. Behavior.Change; any.q.Doctor.Physical; any.q. Physically.Challenged; age.c.5; pandemic. period; any.q.Protein.Vitamin; any.q.Isolation. Comb; any.q.Mental.Comb

**Funding:** This work was supported by a grant from the United Nations Development Program, Bangladesh (UNDP; https://www.undp.org/ bangladesh) to Columbia University. AS, DMA, and TK is supported by the UNDP grant. The content is solely the responsibility of the authors and does not necessarily represent the official views of the funding agency. The funders had no role in the study design, data collection, and analysis, decision to publish, or preparation of the manuscript.

**Competing interests:** The authors have declared that no competing interests exist.

higher odds (95% Confidence Interval, CI: 1.09, 1.22) of death compared to females. Compared to the reference age group (20–24 years olds), the odds ratio of mortality increased monotonically with age, ranging from an odds ratio of 1.35 (95% CI: 1.05, 1.73) for ages 30–34 to an odds ratio of 21.6 (95% CI: 17.08, 27.38) for ages 75–79 year group. For children 0–4 years old the odds of mortality were 3.93 (95% CI: 2.74, 5.64) times higher than 20–24 years olds. Other significant predictors were severe symptoms of COVID-19 such as breathing difficulty, fever, and diarrhea. Patients who were assessed by a physician as having a severe episode of COVID-19 based on the telehealth interview had 12.43 (95% CI: 11.04, 13.99) times higher odds of mortality compared to those assessed to have a mild episode. The finding that the telehealth doctors' assessment of disease severity was highly predictive of subsequent COVID-19 mortality, underscores the feasibility and value of the telehealth services.

## Conclusions

Our findings confirm the universality of certain COVID-19 risk factors—such as gender and age—while highlighting other risk factors that appear to be more (or less) relevant in the context of Bangladesh. These findings on the demographic, socioeconomic, and clinical risk factors for COVID-19 mortality can help guide public health and clinical decision-making. Harnessing the benefits of the telehealth system and optimizing care for those most at risk of mortality, particularly in the context of a LMIC, are the key takeaways from this study.

## Introduction

The COVID-19 pandemic has claimed over 6.45 million lives [1] (as of August 2022) worldwide since the first official reported death in January 2020. Mortality rates and case-fatality ratio for COVID-19 have varied widely across countries and across different waves of the pandemic [2]. While several risk factors for severe disease due to COVID-19, most notably age, have been well-established [3], understanding who is most at risk of hospitalization and death remains an important public health priority. Understanding these risk factors is especially important in the context of low- and middle-income countries (LMICs) where resources are scarce and therefore need to be prioritized to care for those who are at highest risk of adverse outcomes. The relative importance of exposure to demographic, socioeconomic and clinical risk factors on COVID-19 severity and mortality in Bangladesh is not adequately studied. Bangladesh, a lower middle-income country in South Asia with a dense population, has observed 12,136 total cases per million population and 175 deaths per million population during the pandemic [4].

The majority of studies examining risk factors for severe disease and death due to COVID-19 have relied on population-based inferences, and have shown associations with age [5, 6], gender [7], and socioeconomic status [8]. Data linkage between death records and other sources of individual-level data, such as surveys, is available in only a handful of countries [9–14], and rarely in a LMIC. The majority of studies from LMIC are from sub-Saharan Africa [15], while studies from South Asia are mostly cross-sectional, with small patient populations and limited data on potential predictors [16–18]. To date, two studies using small datasets with limited information on socioeconomic factors, symptom severity and preexisting

conditions have examined the risk factors of severe illness and mortality due to COVID-19 in Bangladesh [19].

Here, we use data from 290,488 lab-confirmed COVID-19 patients who participated in a telehealth service in Bangladesh, linked with COVID-19 death data from a national database, to study the risk factors associated with mortality. All patients who participated in the telehealth service between May 17, 2020 and June 14, 2021 were eligible to be included in the study. During this period, there were approximately 807,704 COVID-19 cases and 12,844 COVID-19 attributable deaths recorded in Bangladesh [1]; our study, thus, covers about 36% of all recorded cases in the country during the study period. Mass vaccination was rolled out in Bangladesh on February 21st, 2021 and only a small proportion (<4%) of the Bangladeshi population was vaccinated during our study period ending in June 2021 [20].

The contribution of our study is two-fold. First, understanding the demographic, socioeconomic, and clinical risk factors for COVID-19 mortality can help guide clinical decision-making. These results are particularly relevant in resource-constrained settings where hospital beds and ICU capacity may be limited, or during infection surges when there is an impending shortage in healthcare services. Knowledge of the risk factors can also guide policymakers in targeting non-pharmaceutical interventions toward populations that are most at-risk. Second, our analysis provides support for the usefulness of telehealth services in Bangladesh. The physician's evaluation of patients, based on the interviews through the telehealth service, was highly predictive of eventual mortality; this finding highlights the feasibility of using telehealth services for triaging patients who are the most at-risk.

## Methods

### Study design and data sources

We conducted a prospective cohort study using routinely collected electronic data from PCR positive COVID-19 patients who received telehealth services provided by the Government of Bangladesh between May 17, 2020 and June 14, 2021. Under this program, PCR positive COVID-19 patients received a call from a health information officer (HIO) who confirmed their COVID status and conducted an initial assessment of their health condition. The HIOs then transferred the patients to a physician, who further assessed patients' health status and advised them regarding the next course of their treatment. Based on the physician's assessment, follow-up calls were scheduled. Every patient who agreed to receive the telehealth services received at least one phone call and a follow-up call after 3, 5, or 10 days depending on whether they were assessed as having severe, moderate, or mild symptoms respectively. An additional follow-up call was scheduled for 7, 10, and 14 days after the initial call based on the assessed severity of symptoms.

During the study period, 334,626 patients received telehealth services. Of these, we removed all patients whose age and sex data were not correctly recorded (7823 patients), patients with one or more questions with seemingly incorrectly entered data (103 patients), and patients without mortality information (36,212 patients). Our final study population included 290,488 patients.

Ethical review for this study was sought from The New York State Psychiatric Institute (NYSPI IRB protocol #8173) Institutional Review Board. The board determined that the secondary data analysis of routine patient data was exempt from ethical review and approval as it did not meet the definition of human subject research according to federal guidelines. We also obtained ethical approval from the institutional review board of International Center for Diarrhoeal Disease Research in Bangladesh, (icddr,b IRB protocol PR-23030)

## Outcomes and predictor variables

The outcome of our analysis was a binary indicator of whether a death was recorded for a patient in the study population. The telehealth database recorded deaths when family members reported the death during follow-up calls. We obtained additional death information from the government's COVID-19 death database compiled by the Directorate General of Health Services. The telehealth and mortality database were merged by the authorities. To combine the databases, unique identifier in the death database was created by finding unique combinations of patients' mobile number, age, gender, and district of residence. The same parameters were matched on the telehealth database and the unique identifier was confirmed. This unique ID was later used for merging the two databases for further analysis. Then de-identified data was shared with us for the analysis of risk factors presented in this paper.

We analyzed potential risk factors of COVID-19 mortality based on the questions asked by the physicians during clinical assessment. The telehealth questionnaire captured data on patient demographics, living conditions, pandemic period, pre-existing health conditions, presenting symptoms, patients' self-assessment of health status, physicians' assessment of patients' mental status, and physicians' assessment of patients' overall health condition.

Patients' age was recorded as a continuous variable, which we categorized into 5-year groups, with ages 20–25 treated as the reference category. Use of categorical age variable allowed us to examine non-linear relationship of age with mortality. Five-year age groups were chosen as similar age categories was used in analyses of age effects on COVID-19 mortality in previous studies [21]. We used proxies for determining a patient's socioeconomic status (SES) as we did not have direct measures of income, occupation, education, etc. We categorized a patient as having low SES if they reported that they were unable to isolate at home (indicating crowded living conditions) or did not have a separate bedroom and bathroom for their use in the house.

Depending on the date of the first call to the telehealth service, patients were from pandemic period 1 (May 17, 2020, to September 30, 2020), period 2 (October 1, 2020, to January 31, 2021), period 3 (February 1, 2021, to May 14, 2021), and period 4 (May 14, 2021, to June 14, 2021). The time periods are categorized considering the multiple waves of cases observed, and predominant COVID-19 variants in Bangladesh. The variant Alpha (B.1.1.7) was detected in Bangladesh in December 2020, which continued to be the predominant variant until late 2020 when the variant Beta (B.1.35) emerged. During February and March 2021, the Beta variant became the predominant variant accounting for 90% of all cases. When the Delta (B.1.617.2) variant arrived in Bangladesh in early May 2021, it had become the most prominent variant constituting 68% of the variants circulating in Dhaka city by the end of May 2021 [22].

Physicians ascertained the presence of comorbidities based on patients' self-reporting and then verified these self-reports by asking follow-up questions about specific medications taken for the condition(s) reported. Comorbidities were categorized as diabetes, hypertension, chronic kidney disease, chronic liver disease, chronic respiratory disease (asthma, COPD), and others (thyroid conditions, Alzheimer's disease, cancers, and heart disease). Patients' reports of symptoms including fever, cough, chest pain, loss of taste and smell, headache, weakness, diarrhea, and vomiting were recorded as none, mild, moderate, and severe. Physicians recorded their overall assessment of the patient's physical condition as mild, moderate, and severe, and patients were advised on treatment and follow-up plans accordingly. Measures of mental health include physicians' records of patients' mental condition: normal, stressed, or panicked, and also patients' report of having adequate sleep. Patients also reported their own assessment of the improvement of their health status.

## Statistical analyses

To examine the association between potential risk factors and mortality, we used multiple logistic regression models, with death as the binary outcome. The odds ratios of mortality estimated from the logistic regressions approximates risk ratios given the low incidence of the outcome. We preferred logistic regression over binomial regression because of better model performance. The odds ratios (OR) presented in the manuscript are adjusted odds ratios obtained from the three separate multivariable models. We used separate multivariable models for three sets of predictor variables based on their temporal location in the disease pathway (S1 Fig). This allows us to avoid introducing bias in our analyses by adjusting for variables on the hypothesized causal pathway between a risk factor and mortality. The three models with different sets of predictors are: (1) model 1 that included patient's age, sex, location, period of the pandemic, socioeconomic status, comorbidities, and symptoms; (2) model 2 that included patient's age, sex, patient's mental health status, patient's rating of health improvement, and amount of sleep they were getting; (3) model 3 that included patients age, sex, and physician's rating of a patient's health condition. Missing values for covariates were replaced with a "missing" indicator and included in the multivariable models.

To evaluate which risk factors are most important in the context of clinical decision-making, we used classification and regression tree (CART) models to generate decision trees. The CART is a statistical technique based on recursive partitioning analysis and is well suited for the generation of clinical decision rules [23–25]. Unlike multiple logistic regression, it can handle numerical data that are highly skewed or multimodal and categorical predictors with either an ordinal or nominal structure. The CART involves segregating different values of classification variables through a decision tree composed of progressive binary splits. Every value of each predictor is considered as a potential split, and the optimal split is selected based on the reduction in the residual sum of squares due to a binary split of the data at that tree node. Each parent node produces two child nodes, which can become parent nodes producing additional child nodes. This process continues with tree building and pruning until the tree fits without overfitting the information contained in the data set. We used the R rpart function from the rpart package [26], with method argument "class". A node needed to contain at least 20 observations for a split to be attempted. Any split that did not decrease the overall lack of fit by a factor of 0.001 was not considered. 10 cross-validations were run.

In our data, far more patients survived than deceased. Due to the imbalance, the model tends to focus on the prevalent class and to ignore the rare events, and the scarcity of data leads to poor estimates of the model's accuracy. We generated artificial balanced samples according to a smoothed bootstrap approach for aiding estimation and accuracy evaluation of a binary classifier in the presence of a rare class using the R ROSE package [27, 28]. All analyses were performed using R version 4.0.

## Results

Our final study population included 290,488 patients, representing around 36% of total COVID-19 cases in the country during that period. 6,951 deaths were recorded among the patients included in our analyses (2.4% of study population). The characteristics of the study population are shown in Table 1. The majority (68%) of the patients were men and resided in Dhaka (54%). Of the 191,775 cases assessed by physicians, most were mild; only 16% were moderate and 1.3% were severe cases.

Figs 1 and 2 show the results of the multivariate regression for model 1. Men, very young children, and older patients had a higher risk of mortality from COVID-19 compared to the rest of the study population. Males had 1.15 times higher odds (95% CI: 1.09, 1.22) of death

**Table 1. Characteristics of the study population.**

| | Overall (N = 290,488) | | No death (N = 283,537) | | Death (N = 6,951) | |
|---|---|---|---|---|---|---|
| | n | % or Mean (SD) | n | % or Mean (SD) | n | % or Mean (SD) |
| **Socio-demographic factors** | | | | | | |
| Age | | | | | | |
| 0–5 | 2,029 | 0.7% | 1,979 | 0.7% | 50 | 0.7% |
| 5–10 | 1,944 | 0.7% | 1,928 | 0.7% | 16 | 0.2% |
| 10–15 | 3,292 | 1.1% | 3,261 | 1.2% | 31 | 0.4% |
| 15–20 | 9,701 | 3.3% | 9,629 | 3.4% | 72 | 1.0% |
| 20–24 | 17,697 | 6.1% | 17,611 | 6.2% | 86 | 1.2% |
| 25–30 | 36,061 | 12.4% | 35,853 | 12.6% | 208 | 3.0% |
| 30–35 | 36,195 | 12.5% | 35,971 | 12.7% | 224 | 3.2% |
| 35–40 | 38,854 | 13.4% | 38,521 | 13.6% | 333 | 4.8% |
| 40–45 | 28,543 | 9.8% | 28,247 | 10.0% | 296 | 4.3% |
| 45–50 | 27,956 | 9.6% | 27,497 | 9.7% | 459 | 6.6% |
| 50–55 | 24,424 | 8.0% | 23,837 | 8.4% | 587 | 8.4% |
| 55–60 | 22,720 | 7.8% | 21,891 | 7.7% | 829 | 11.9% |
| 60–65 | 14,645 | 5.0% | 13,871 | 4.9% | 774 | 11.1% |
| 65–70 | 13,004 | 4.5% | 11,870 | 4.2% | 1,134 | 16.3% |
| 70–75 | 6,733 | 2.3% | 6,015 | 2.1% | 718 | 10.3% |
| 75–80 | 3,676 | 1.3% | 3,138 | 1.1% | 538 | 7.7% |
| 80+ | 3,014 | 1.0% | 2,418 | 0.9% | 596 | 8.6% |
| Sex | | | | | | |
| Female | 91,645 | 31.5% | 89,739 | 31.6% | 1,906 | 27.4% |
| Male | 198,843 | 68.2% | 193,798 | 68.4% | 5,045 | 72.6% |
| Region | | | | | | |
| Dhaka | 150,610 | 51.8% | 147,418 | 52.0% | 3,192 | 45.9% |
| Chittagong | 39,412 | 13.6% | 38,426 | 13.6% | 986 | 14.2% |
| Mymensingh | 5,814 | 2.0% | 5,673 | 2.0% | 141 | 2.0% |
| Barisal | 9,244 | 3.2% | 9,002 | 3.2% | 242 | 3.5% |
| Sylhet | 7,387 | 2.5% | 7,134 | 2.5% | 253 | 3.6% |
| Khulna | 15,474 | 5.3% | 15,082 | 5.3% | 392 | 5.6% |
| Rajshahi | 15,357 | 5.3% | 15,048 | 5.3% | 309 | 4.4% |
| Rangpur | 9,662 | 3.3% | 9,466 | 3.3% | 196 | 2.8% |
| Presence of a Caregiver | | | | | | |
| No | 35,398 | 12.2% | 34,862 | 12.3% | 536 | 7.7% |
| Yes | 146,549 | 50.4% | 144,732 | 51.0% | 1,817 | 26.1% |
| Missing | 108,541 | 37.4% | 103,943 | 36.7% | 4,598 | 66.1% |
| Socioeconomic status | | | | | | |
| Living conditions not crowded | 23,247 | 8.0% | 22,828 | 8.1% | 419 | 6.0% |
| Crowded living conditions | 155,097 | 53.4% | 153,766 | 54.2% | 1,331 | 19.1% |
| **Pandemic Period** | | | | | | |
| 2020-05-17 to 2020-09-30 | 152,532 | 52.5% | 149,001 | 52.6% | 3,531 | 50.8% |
| 2020-10-01 to 2021-01-31 | 96,084 | 33.1% | 93,832 | 33.1% | 2,252 | 32.4% |
| 2021-02-01 to 2021-05-14 | 40,395 | 13.9% | 39,531 | 13.9% | 864 | 12.4% |
| 2021-05-15 to 2021-06-15 | 1,477 | 0.5% | 1,173 | 0.4% | 304 | 4.4% |
| **Change in behavior after Covid Diagnosis** | | | | | | |
| No | 170,219 | 58.6% | 168,167 | 59.3% | 2,052 | 29.5% |
| Yes | 16,971 | 5.8% | 16,594 | 5.9% | 377 | 5.4% |

*(Continued)*

**Table 1.** (Continued)

| | Overall (N = 290,488) | | No death (N = 283,537) | | Death (N = 6,951) | |
|---|---|---|---|---|---|---|
| | n | % or Mean (SD) | n | % or Mean (SD) | n | % or Mean (SD) |
| Missing | 103,298 | 35.6% | 98,776 | 34.8% | 4,522 | 65.1% |
| **Rules at home after Covid diagnosis** | | | | | | |
| Distance with Family | | | | | | |
| No | 26,737 | 9.2% | 26,478 | 9.3% | 259 | 3.7% |
| Yes | 139,039 | 47.9% | 137,927 | 48.6% | 1,112 | 16.0% |
| Missing | 124,712 | 42.9% | 119,132 | 42.0% | 5,580 | 80.3% |
| Masking | | | | | | |
| No | 47,457 | 16.3% | 46,991 | 16.6% | 466 | 6.7% |
| Yes | 118,319 | 40.7% | 117,414 | 41.4% | 905 | 13.0% |
| Missing | 124,712 | 42.9% | 119,132 | 42.0% | 5,580 | 80.3% |
| Isolation | | | | | | |
| No | 23,247 | 8.0% | 22,828 | 8.1% | 419 | 6.0% |
| Yes | 155,097 | 53.4% | 153,766 | 54.2% | 1,331 | 19.1% |
| Missing | 112,144 | 38.6% | 106,943 | 37.7% | 5,201 | 74.8% |
| **Self-rated Patient Health** | | | | | | |
| Health improvement | | | | | | |
| No | 12,710 | 4.4% | 12,032 | 4.2% | 678 | 9.8% |
| Yes | 176,272 | 60.7% | 174,411 | 61.5% | 1,861 | 26.8% |
| Missing | 101,506 | 34.9% | 97,094 | 34.2% | 4,412 | 63.5% |
| Enough Sleep | | | | | | |
| No | 21,941 | 7.6% | 21,290 | 7.5% | 651 | 9.4% |
| Yes | 167,712 | 57.7% | 165,883 | 58.5% | 1,829 | 26.3% |
| Missing | 100,835 | 34.7% | 96,364 | 34.0% | 4,471 | 64.3% |
| **Comorbidities** | | | | | | |
| Chronic respiratory illness | | | | | | |
| No | 179,571 | 61.8% | 176,816 | 62.4% | 2,755 | 39.6% |
| Yes | 9,959 | 3.4% | 9,650 | 3.4% | 309 | 4.4% |
| Missing | 100,958 | 34.8% | 97,071 | 34.2% | 3,887 | 55.9% |
| Diabetes | | | | | | |
| No | 152,757 | 52.6% | 151,062 | 53.3% | 1,695 | 24.4% |
| Yes | 36,773 | 12.7% | 35,404 | 12.5% | 1,369 | 19.7% |
| Missing | 100,958 | 34.8% | 97,071 | 34.2% | 3,887 | 55.9% |
| High Blood Pressure | | | | | | |
| No | 153,222 | 52.7% | 151,512 | 53.4% | 1,710 | 24.6% |
| Yes | 36,308 | 12.5% | 34,954 | 12.3% | 1,354 | 19.5% |
| Missing | 100,958 | 34.8% | 97,071 | 34.2% | 3,887 | 55.9% |
| Kidney Disease | | | | | | |
| No | 186,313 | 64.1% | 183,617 | 64.8% | 2,696 | 38.8% |
| Yes | 3,217 | 1.1% | 2,849 | 1.0% | 368 | 5.3% |
| Missing | 100,958 | 34.8% | 97,071 | 34.2% | 3,887 | 55.9% |
| Chronic Liver Disease | | | | | | |
| No | 188,294 | 64.8% | 185,296 | 65.4% | 2,998 | 43.1% |
| Yes | 1,236 | 0.4% | 1,170 | 0.4% | 66 | 0.9% |
| Missing | 100,958 | 34.8% | 97,071 | 34.2% | 3,887 | 55.9% |
| Other | | | | | | |
| No | 176,806 | 60.9% | 174,241 | 61.5% | 2,565 | 36.9% |

(Continued)

**Table 1.** (Continued)

| | Overall (N = 290,488) | | No death (N = 283,537) | | Death (N = 6,951) | |
|---|---|---|---|---|---|---|
| | n | % or Mean (SD) | n | % or Mean (SD) | n | % or Mean (SD) |
| Yes | 12,724 | 4.4% | 12,225 | 4.3% | 499 | 7.2% |
| Missing | 100,958 | 34.8% | 97,071 | 34.2% | 3,887 | 55.9% |
| **Physician Assessment** | | | | | | |
| Mental Health Status | | | | | | |
| No | 161,312 | 55.5% | 159,742 | 56.3% | 1,570 | 22.6% |
| Yes | 24,077 | 8.3% | 23,252 | 8.2% | 825 | 11.9% |
| Missing | 105,099 | 36.2% | 100,543 | 35.5% | 4,556 | 65.5% |
| Physically Challenged | | | | | | |
| No | 181,279 | 62.4% | 178,971 | 63.1% | 2,308 | 33.2% |
| Yes | 2,813 | 1.0% | 2,635 | 0.9% | 178 | 2.6% |
| Missing | 106,396 | 36.6% | 101,931 | 35.9% | 4,465 | 64.2% |
| Protein and Vitamin | | | | | | |
| Yes | 184,452 | 63.5% | 182,259 | 64.3% | 2,193 | 31.5% |
| Missing | 106,036 | 36.5% | 101,278 | 35.7% | 4,758 | 68.5% |
| **Presenting Symptoms** | | | | | | |
| Body Ache | | | | | | |
| No | 186,889 | 64.3% | 184,461 | 65.1% | 2,428 | 34.9% |
| Mild | 8,608 | 3.0% | 8,488 | 3.0% | 120 | 1.7% |
| Moderate | 5,143 | 1.8% | 5,003 | 1.8% | 140 | 2.0% |
| Severe | 670 | 0.2% | 637 | 0.2% | 33 | 0.5% |
| Missing | 89,178 | 30.7% | 84,948 | 30.0% | 4,230 | 60.9% |
| Breathing Difficulty | | | | | | |
| No | 182,074 | 62.7% | 180,520 | 63.7% | 1,554 | 22.4% |
| Mild | 11,589 | 4.0% | 11,296 | 4.0% | 293 | 4.2% |
| Moderate | 5,793 | 2.0% | 5,367 | 1.9% | 426 | 6.1% |
| Severe | 1,854 | 0.6% | 1,406 | 0.5% | 448 | 6.4% |
| Missing | 89,178 | 30.7% | 84,948 | 30.0% | 4,230 | 60.9% |
| Cough | | | | | | |
| No | 140,137 | 48.2% | 138,440 | 48.8% | 1,697 | 24.4% |
| Mild | 44,784 | 15.4% | 44,286 | 15.6% | 498 | 7.2% |
| Moderate | 15,116 | 5.2% | 14,714 | 5.2% | 402 | 5.8% |
| Severe | 1,273 | 0.4% | 1,149 | 0.4% | 124 | 1.8% |
| Missing | 89,178 | 30.7% | 84,948 | 30.0% | 4,230 | 60.9% |
| Diarrhea | | | | | | |
| No | 196,327 | 67.6% | 193,711 | 68.3% | 2,616 | 37.6% |
| Mild | 3,355 | 1.2% | 3,310 | 1.2% | 45 | 0.6% |
| Moderate | 1,499 | 0.5% | 1,456 | 0.5% | 43 | 0.6% |
| Severe | 129 | 0.0% | 112 | 0.0% | 17 | 0.2% |
| Missing | 89,178 | 30.7% | 84,948 | 30.0% | 4,230 | 60.9% |
| Fever | | | | | | |
| No | 179,178 | 61.7% | 177,004 | 62.4% | 2,174 | 31.3% |
| Mild | 16,965 | 5.8% | 16,661 | 5.9% | 304 | 4.4% |
| Moderate | 4,627 | 1.6% | 4,448 | 1.6% | 179 | 2.6% |
| Severe | 540 | 0.2% | 476 | 0.2% | 64 | 0.9% |
| Missing | 89,178 | 30.7% | 84,948 | 30.0% | 4,230 | 60.9% |
| Headache | | | | | | |

(*Continued*)

**Table 1.** (Continued)

| | Overall (N = 290,488) | | No death (N = 283,537) | | Death (N = 6,951) | |
|---|---|---|---|---|---|---|
| | **n** | **% or Mean (SD)** | **n** | **% or Mean (SD)** | **n** | **% or Mean (SD)** |
| No | 190,814 | 65.7% | 188,249 | 66.4% | 2,565 | 36.9% |
| Mild | 7,886 | 2.7% | 7,812 | 2.8% | 74 | 1.1% |
| Moderate | 2,324 | 0.8% | 2,265 | 0.8% | 59 | 0.8% |
| Severe | 286 | 0.1% | 263 | 0.1% | 23 | 0.3% |
| Missing | 89,178 | 30.7% | 84,948 | 30.0% | 4,230 | 60.9% |
| Loss of taste and smell | | | | | | |
| No | 159,284 | 54.8% | 157,196 | 55.4% | 2,088 | 30.0% |
| Mild | 24,578 | 8.5% | 24,283 | 8.6% | 295 | 4.2% |
| Moderate | 14,444 | 5.0% | 14,178 | 5.0% | 266 | 3.8% |
| Severe | 3,004 | 1.0% | 2,932 | 1.0% | 72 | 1.0% |
| Missing | 89,178 | 30.7% | 84,948 | 30.0% | 4,230 | 60.9% |
| Weakness | | | | | | |
| No | 145,095 | 49.9% | 143,423 | 50.6% | 1,672 | 24.1% |
| Moderate | 20,786 | 7.2% | 20,293 | 7.2% | 493 | 7.1% |
| Mild | 32,037 | 11.0% | 31,718 | 11.2% | 319 | 4.6% |
| Severe | 3,392 | 1.2% | 3,155 | 1.1% | 237 | 3.4% |
| Missing | 89,178 | 30.7% | 84,948 | 30.0% | 4,230 | 60.9% |
| **History of Traveling Abroad** | | | | | | |
| No | 173,376 | 59.7% | 171,061 | 60.3% | 2,315 | 33.3% |
| Yes | 10,310 | 3.5% | 10,158 | 3.6% | 152 | 2.2% |
| Missing | 106,942 | 36.8% | 102,318 | 36.1% | 4,484 | 64.5% |
| **Hospitalization** | | | | | | |
| No | 262,823 | 90.5% | 258,514 | 91.2% | 4,309 | 62.0% |
| Yes | 27,567 | 9.5% | 25,023 | 8.8% | 2,544 | 36.6% |
| Missing | 98 | 0.0% | 0 | 0.0% | 98 | 1.4% |

Descriptive statistics (number and percentage) for the overall sample of all patients, patients who survived, and patients who died.

compared to females. Compared to the reference age group (20–24 years olds), the odds ratio of mortality increased monotonically with age, ranging from an odds ratio of 1.35 (95% CI: 1.05, 1.73) for ages 30–34 to an odds ratio of 21.6 (95% CI: 17.08, 27.38) for ages 75–79 year group. For children 0–4 years old the odds of mortality was 3.9 (95% CI: 2.74, 5.64) times higher than 20–24 years olds.

We observed variation in mortality risk by socioeconomic status, region of the country, and the period of the pandemic. People from low SES (as defined by our proxy measure) had 1.76 (95% CI: 1.56, 1.98) times higher odds of mortality compared to people from low SES backgrounds. Compared to Dhaka, patients from 3 divisions had higher odds of mortality, with the highest odds among the patients from Sylhet (OR: 1.62, 95% CI: 1.40, 1.87). Compared to the first period of the pandemic in our study, the odds ratio of mortality was elevated in the last period (May 14, 2021, to June 14, 2021), although the proportion of cases from period 3 was low (6.7%) due to the cutoff date for the available data.

Compared to people without any comorbidities, the odds of mortality was 2.34 times (95% CI: 2.04, 2.68) higher in patients with chronic kidney disease, and 2.08 (95% CI: 1.57, 2.74) times higher in patients with chronic liver disease. Patients with hypertension (high blood pressure) also had a slightly elevated risk of mortality (OR 1.13, 95% CI: 1.04, 1.23). However, the prevalence of chronic kidney and liver disease was low compared to the prevalence of

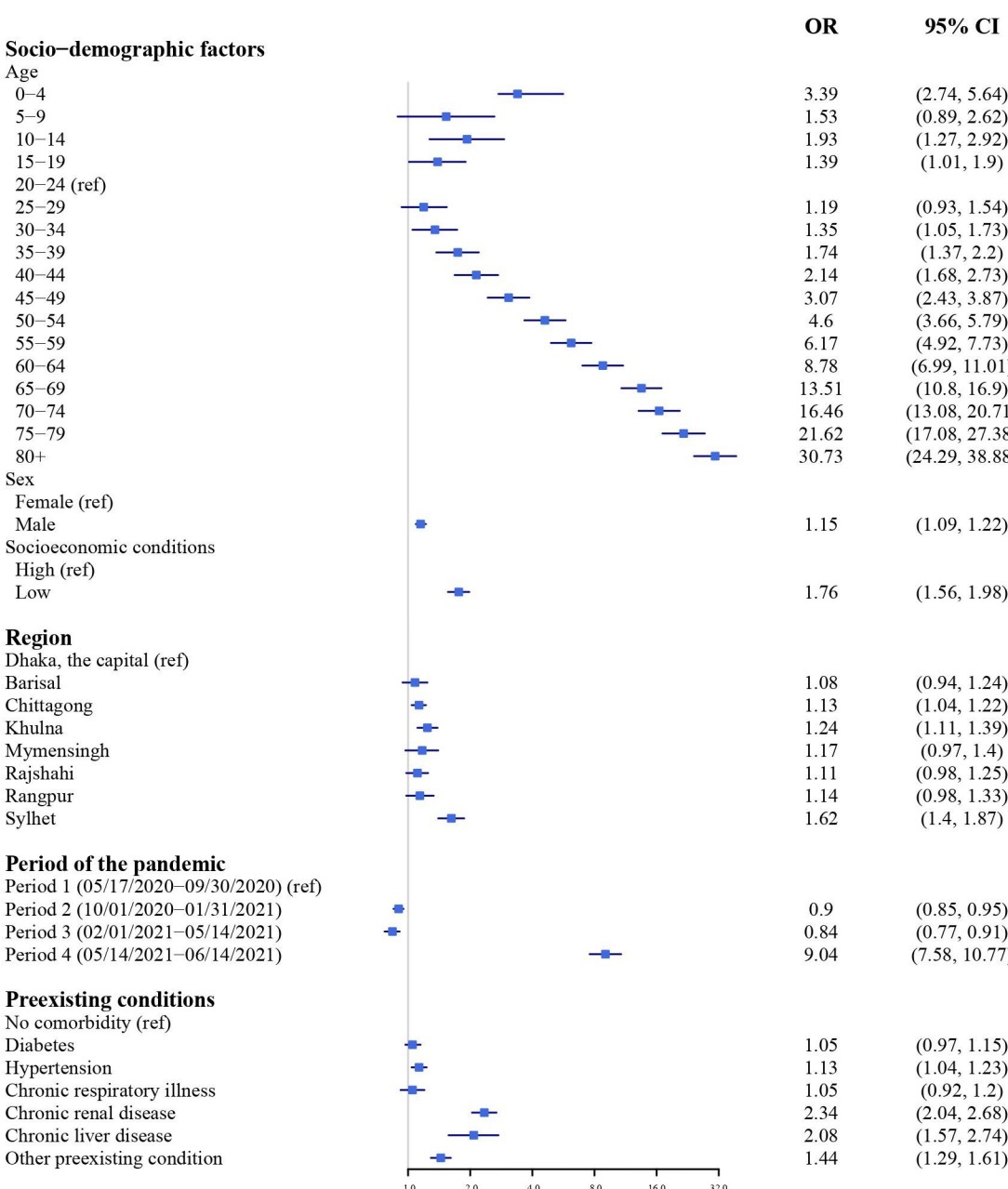

**Fig 1. Estimated odds ratios and 95% confidence intervals for mortality among COVID-19 patients from the multivariable logistic regression model with patient's age, sex, location, period of the pandemic, socioeconomic status, comorbidities, and symptoms as the predictors (model 1).** Results are shown here for patient demographics, region, period, and the pandemic and pre-existing conditions. Full model results are shown in Table 2.

hypertension, indicating a higher attributable risk of mortality due to hypertension (Fig 2). Diabetes did not have a significant association with mortality after adjustment for other comorbidities and presenting symptoms. Mortality risk was also not elevated among patients with chronic respiratory illness (asthma and COPD).

Among the presenting symptoms, breathing difficulty, fever, diarrhea, and body ache were significantly associated with mortality (Table 2, model 1). The risk of mortality increased with

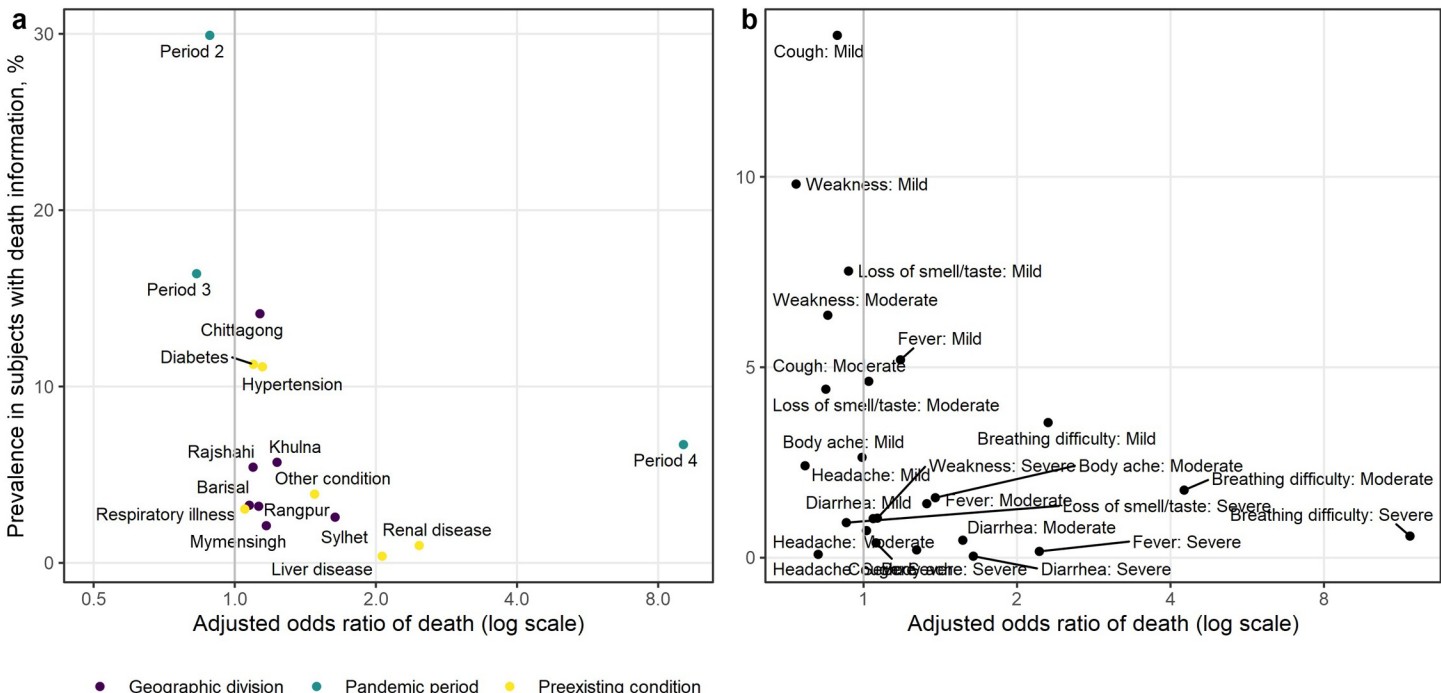

**Fig 2. Prevalence in the study population versus the estimated odds ratios from model 1.** Risk factors in panel a include patients' location, the pandemic period, and pre-existing conditions; Risk factors in panel b include presenting symptoms reported by the patients.

severity of breathing difficulty with 1.95 (95% CI: 1.70, 2.24), 3.25 (95% CI: 2.85, 3.71), and 8.23 (95% CI: 7.05, 9.60) times higher odds of death among patients with mild, moderate, and severe breathing difficulty, respectively. Similarly, the odds of mortality were 1.20 (CI: 1.05, 1.37), 1.34 (1.12, 1.60), and 2.16 (1.51, 3.10), times higher among patients reporting mild, moderate, and severe fever. However, the proportions of patients reporting severe breathing difficulty and severe fever were low (Fig 2). There was no association of cough, vomiting, and loss of taste and smell with mortality.

We also estimated the association between COVID-19 mortality and the patient's mental health status and assessment of their health (Table 3, model 2) and the physician's assessment of the patient's health (Table 4, model 3) while controlling for age and gender. Patients who reported that their health was improving and reported having enough sleep had lower odds of mortality; patient's whose mental health state was considered normal by physician's also had lower odds of mortality (Table 3). Physician's assessment of a patient's health status was highly predictive of mortality; patients who were assessed by a physician as having a severe episode of COVID-19 based on the telehealth interview had 12.43 (CI: 11.04, 13.99) times higher odds of mortality compared to those assessed to have a mild episode (Table 4).

Finally, Fig 3 shows the classification tree for death among COVID-19 patients using age, gender, pre-existing conditions, and COVID-19 symptoms. In our main model (excluding physician's assessment of the patient's condition), age and the presence of breathing difficulty played an important role, appearing in numerous nodes across the tree. For example, older patients over the age of 55 with breathing difficulties showed high risk of death (13% mortality rate within the group), while younger people below the age of 55 and without breathing difficulties and without pre-existing conditions showed very low risk (1% mortality rate). As expected, based on the results of model 3 above, the physician's health assessment appears important when included in the model (S2 Fig). For example, older patients over the age of 55

**Table 2. Model 1: Comorbidities, presenting symptoms, SES, pandemic period, location, age, and sex.**

| term | estimate | Adjusted OR (95% CI) |
|---|---|---|
| **Socio-demographic factors** | | |
| Age | | |
| 20–24 (ref) | | |
| Intercept | -6.43 | 0.00 (0.00, 0.00) |
| 0–4 | 1.37 | 3.93 (2.74, 5.63) |
| 5–9 | 0.42 | 1.53 (0.89, 2.62) |
| 10–14 | 0.66 | 1.93 (1.28, 2.92) |
| 15–19 | 0.33 | 1.39 (1.01, 1.90) |
| 25–29 | 0.18 | 1.19 (0.93, 1.54) |
| 30–34 | 0.30 | 1.35 (1.05, 1.73) |
| 35–39 | 0.55 | 1.74 (1.37, 2.20) |
| 40–44 | 0.76 | 2.14 (1.68, 2.73) |
| 45–49 | 1.12 | 3.07 (2.43, 3.87) |
| 50–54 | 1.53 | 4.60 (3.67, 5.79) |
| 55–59 | 1.82 | 6.17 (4.92, 7.73) |
| 60–64 | 2.17 | 8.78 (6.99, 11.01) |
| 65–69 | 2.60 | 13.51 (10.80, 16.90) |
| 70–75 | 2.80 | 16.46 (13.09, 20.71) |
| 75–79 | 3.07 | 21.62 (17.08, 27.38) |
| 80+ | 3.43 | 30.73 (24.29, 38.88) |
| Sex | | |
| Female (ref) | | |
| Male | 0.14 | 1.15 (1.09, 1.22) |
| Region | | |
| Dhaka (ref) | | |
| Missing | -0.18 | 0.84 (0.78, 0.90) |
| Barisal | 0.08 | 1.08 (0.94, 1.24) |
| Chittagong | 0.12 | 1.13 (1.04, 1.23) |
| Mymensingh | 0.15 | 1.17 (0.97, 1.40) |
| Khulna | 0.21 | 1.24 (1.11, 1.39) |
| Rajshahi | 0.10 | 1.09 (0.98, 1.25) |
| Rangpur | 0.13 | 1.14 (0.98, 1.33) |
| Sylhet | 0.48 | 1.62 (1.40, 1.87) |
| **First Call Date (Pandemic Period)** | | |
| 2020-05-17 to 2020-09-30 (ref) | | |
| 2020-10-01 to 2021-01-31 | -0.11 | 0.89 (0.85, 0.95) |
| 2021-02-01 to 2021-05-14 | -0.18 | 0.84 (0.77, 0.91) |
| 2021-05-15 to 2021-06-15 | 2.20 | 9.04 (7.60, 10.77) |
| **Socioeconomic status** | | |
| Living conditions not crowded (ref) | | |
| Crowded living conditions | 0.56 | 1.76 (1.56, 1.98) |
| Missing | 0.80 | 2.24 (2.03, 2.46) |
| **Comorbidities** | | |
| Missing | -1.14 | 0.32 (0.28, 0.36) |
| Diabetes | 0.05 | 1.06 (0.97, 1.15) |
| Chronic Respiratory Illness | 0.05 | 1.05 (0.92, 1.23) |
| High Blood Pressure | 0.12 | 1.13 (1.04, 1.23) |

(*Continued*)

**Table 2.** (Continued)

| term | estimate | Adjusted OR (95% CI) |
|---|---|---|
| Kidney Disease | 0.85 | 2.33 (2.04, 2.67) |
| Chronic Liver Disease | 0.73 | 2.08 (1.58, 2.74) |
| Other | 0.37 | 1.44 (1.29, 1.61) |
| **Presenting Symptoms** | | |
| Fever | | |
| No (ref) | | |
| Missing | 2.20 | 9.04 (7.86, 10.40) |
| Mild | 0.18 | 1.20 (1.05, 1.37) |
| Moderate | 0.29 | 1.34 (1.12, 1.60) |
| Severe | 0.77 | 2.16 (1.51, 3.10) |
| Breathing Difficulty | | |
| No (ref) | | |
| Mild | 0.67 | 1.95 (1.70, 2.24) |
| Moderate | 1.18 | 3.25 (2.85, 3.71) |
| Severe | 2.11 | 8.23 (7.05, 9.60) |
| Headache | | |
| No (ref) | | |
| Mild | -0.22 | 0.79 (0.62, 1.03) |
| Moderate | 0.08 | 1.08 (0.80, 1.48) |
| Severe | -0.17 | 0.85 (0.46, 1.55) |
| Weakness | | |
| No (ref) | | |
| Moderate | -0.18 | 0.84 (0.74, 0.95) |
| Mild | -0.31 | 0.73 (0.64, 0.83) |
| Severe | 0.02 | 1.02 (0.84, 1.23) |
| Diarrhea | | |
| No (ref) | | |
| Mild | 0.06 | 1.06 (0.77, 1.46) |
| Moderate | 0.47 | 1.60 (1.14, 2.25) |
| Severe | 0.55 | 1.73 (0.85, 3.51) |
| Body Ache | | |
| Mild | 0.01 | 1.01 (0.82, 1.23) |
| Moderate | 0.35 | 1.41 (1.15, 1.74) |
| Severe | 0.28 | 1.33 (0.82, 2.17) |
| Loss of Taste and Smell | | |
| No (ref) | | |
| Mild | -0.03 | 0.97 (0.85, 1.12) |
| Moderate | -0.14 | 0.87 (0.74, 1.01) |
| Severe | -0.05 | 0.96 (0.71, 1.29) |
| Cough | | |
| No (ref) | | |
| Mild | -0.09 | 0.91 (0.82, 1.02) |
| Moderate | 0.05 | 1.05 (0.92, 1.20) |
| Severe | 0.05 | 1.05 (0.80, 1.36) |

Estimated coefficients, odds ratios and 95% confidence intervals for mortality among COVID-19 patients from the multivariable logistic regression model with patient's age, sex, location, period of the pandemic, socioeconomic status, comorbidities, and symptoms as the predictors (model 1).

**Table 3. Model 2: Self-rated health and mental health variables, age, and sex.**

| term | estimate | Adjusted OR (95% CI) |
|---|---|---|
| **Self-rated Health** | | |
| Health Improvement | | |
| No (ref) | | |
| Yes | -1.23 | 0.29 (0.27, 0.32) |
| Missing | -0.77 | 0.46 (0.39, 0.55) |
| Mental Health | | |
| No (ref) | | |
| Yes | 0.71 | 2.04 (1.86, 2.24) |
| Missing | 0.41 | 1.50 (1.28, 1.76) |
| Enough Sleep | | |
| No (ref) | | |
| Yes | -0.38 | 0.69 (0.62, 0.76) |
| Missing | 0.35 | 1.42 (1.16, 1.73) |

Estimated coefficients, odds ratios and 95% confidence intervals for mortality among COVID-19 patients from the multivariable logistic regression model with patient's age, sex, self-rated health measures (model 2).

but without any breathing difficulties had a high risk of mortality if the physician's assessment was severe (19% mortality rate). Overall, these results highlight risk factors that are most pertinent in the context of clinical decision making and triaging resources towards patients most at risk of mortality.

## Discussion

We conducted a prospective cohort study to examine the risk factors of COVID-19 mortality among 290,488 PCR-confirmed patients from Bangladesh. Results of our analyses show that the risk of dying from COVID-19 among infected patients in Bangladesh was associated with several demographic, socioeconomic, and clinical risk factors. Males, the very young and the elderly, patients from low SES backgrounds, patients with chronic kidney and liver disease, and patients with symptoms indicating severe illness such as breathing difficulty, fever, and diarrhea had a higher risk of mortality compared to other COVID-19 patients. Additionally, we show that a physician's assessment of a patient's health status, based on the telehealth interview, was highly predictive of eventual mortality due to COVID-19. Based on the association of risk factors with mortality in Bangladeshi population, we developed a decision tree to support clinical decision making and referral of COVID-19 patients

**Table 4. Model 3: Physician assessment, age, and sex.**

| term | estimate | Adjusted OR (95% CI) |
|---|---|---|
| **Physician Assessment** | | |
| Mild (ref) | | |
| Severe | 2.52 | 12.43 (11.04, 13.99) |
| Moderate | 0.65 | 1.92 (1.75, 2.10) |
| Missing | 1.49 | 4.44 (4.17, 4.73) |

Estimated coefficients, odds ratios and 95% confidence intervals for mortality among COVID-19 patients from the multivariable logistic regression model with patient's age, sex, and physician's assessment (model 3).

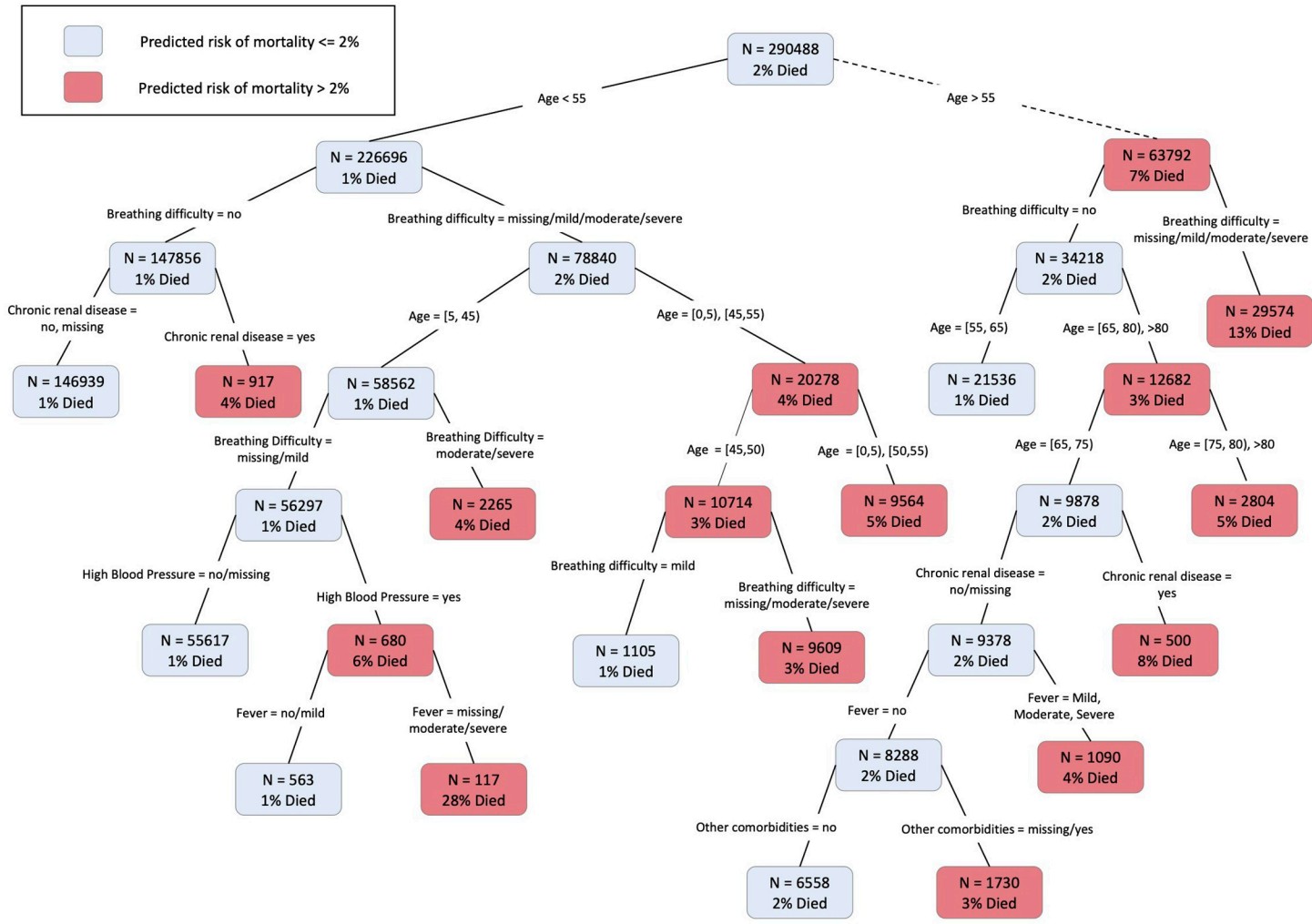

**Fig 3. Classification and regression tree for mortality among COVID-19 patients using age, gender, comorbidities, and presenting symptoms as predictors.** N represents the number of patients in the original data (as opposed to the resampled data used to create the tree). Blue indicates the predicted risk of mortality less than or equal to 2%; red indicates the predicted risk of mortality greater than 2% in the data.

Our findings confirm the universality of certain COVID-19 risk factors—such as gender and age [29]—while highlighting other risk factors that appear to be more (or less) relevant in the Bangladeshi context. In line with previous studies, we find a higher risk of mortality among males [30, 31]. Our finding of a monotonic increase of mortality risk with increasing age is also consistent with prior reports [32–34]. Interestingly, the odds of mortality were relatively high below age 65 (the most common cutoff for determining "high-risk" status for vaccine prioritization, etc.), suggesting that in the LMIC context, mortality the risk may start to increase sharply at an earlier age than observed in high-income settings. We also observed elevated odds of mortality among very young children compared to young adults, which may be due to inflammatory multisystem conditions [35, 36]. The higher mortality among very young children, however, could also be explained by limited testing among this age group resulting in the detection of cases with only severe symptoms.

We observed a significantly elevated risk of mortality associated with chronic renal disease, chronic liver disease, and hypertension, but not diabetes and chronic respiratory illness. Meta-

analyses of prior studies have reported a similar magnitude of risk elevation for hypertension and a larger magnitude of elevation for chronic renal disease and liver disease [37]. Consistent with prior studies, we also found risk elevation among patients with cardiovascular disease and cancer which we grouped into "other" preexisting conditions [38].

Interestingly, we did not find a significant association between COVID-19 mortality with diabetes and chronic respiratory illnesses. The findings on the association between these comorbidities and COVID-19 mortality have been inconclusive in prior literature, with some studies showing a positive association [39]. Both diabetes and hypertension are correlated with old age, therefore residual confounding in previous studies reporting a significant association is plausible. On the other hand, the null results observed in our data could be due to non-differential misclassification. Given the limited access to health care for chronic disease detection in Bangladesh, many COVID-19 patients included in the study may not have been aware of their diabetes and hypertension status [40].

Our study is one of the handful of studies from LMIC to report significantly higher odds of mortality among patients from low SES backgrounds. Previous work has shown that a higher risk of COVID-19 infection and mortality is associated with a higher social vulnerability index [41], income inequality [42], low education [43], immigrant status [44], and black and Hispanic ethnicity ([45]), but the majority of these studies have been conducted in high-income settings. The higher mortality among people from low SES backgrounds may be caused by a combination of factors including poor nutritional status and inadequate management of comorbid conditions, delayed presentation at healthcare facilities, and lack of access to quality care [46, 47]. While we do not have a direct measure of the quality of care, our observation of higher mortality in areas outside Dhaka could be indicative of a lack of access to high quality clinical care in those areas. In addition, lower mortality in Dhaka could also be explained, in part, due to greater access to testing facilities leading to the detection of a greater number of mild and symptomatic cases. The proxy of SES in our study—not having a separate bedroom or a bathroom—is a crude measure and these findings need to be evaluated in future work with more proximal measures of SES such as income, education, and occupational categories.

We find higher odds of mortality among patients who were infected during period 4 (after 5/15/2021). This coincides with the circulation of the Delta variant in Bangladesh. Extending this analysis to cover the duration of the Delta wave—to assess possible heightened mortality risk due to the Delta variant—will be a key direction for future work.

Finally, we present a decision tree highlighting the risk factors that we identified (age older than 55 years, presence of breathing difficulty, and male gender) as most pertinent in the context of clinical decision-making and triaging resources towards patients most at risk of severe COVID-19 and COVID-19 mortality in Bangladesh. Decision trees consolidate our knowledge of the factors that lead to and determine the severity of disease and translate it into clinically actionable items [48]. The decision tree we present fills the gap for a handy decision-making algorithm that can be used by first-level health workers for the initial assessments of COVID-19 patients during telehealth visits. The decision tree presented in our paper can also support clinicians in different parts of Bangladesh to make quantitatively prudent decisions to refer critical COVID-19 patients to health centers equipped to manage severe cases.

Our study has several limitations. First, although all COVID-19 patients in the country were eligible for the telehealth program, there are likely selection biases in the sample as testing was not universal and not all patients may have had access to a telephone, answered the telehealth calls, or agreed to participate in the service. However, mobile phone ownership is high in Bangladesh (178.61 million subscribers as of 2021). According to recent reports, 56% of the population has a personal mobile phone and almost every household has at least one mobile phone [49, 50]. Therefore the patients eligible for this study are not likely to be substantially

different from the general population. Second, we lacked data on smoking status, obesity, immunosuppression, and clinical and laboratory markers of a patient's health condition—all of which may be associated with COVID-19 severity and mortality. Third, telehealth physicians rated the severity of the patient's condition based on patient's self-reporting of symptoms and preexisting conditions. Patients' self-reports and telehealth physicians' assessments were not validated against in-person clinical assessment and therefore the accuracy of the assessment cannot be established. Finally, there is also a significant amount of missing data for the outcome (S1 Table) and several of the risk factors analyzed in this study (Table 1). It is likely that the missingness of risk factor data was associated with the severity of the disease, given severely ill and hospitalized patients were less likely to complete phone assessments and respond to follow-up calls. Therefore, the proportion of missing risk factor data among severely ill patients is likely to bias our estimates toward the null. We partially addressed the issue of missing outcome data by merging the death database records with the telehealth records to capture a much larger proportion of all confirmed COVID-19 deaths in the country over that period. Our results are generalizable to unvaccinated patients in Bangladesh and other low resource settings, as most of the Bangladeshi population was unvaccinated during the period of telehealth services. Although, current vaccination coverage is high in Bangladesh, coverage among vulnerable poor and elderly in rural areas are still low and the provision of boosters is inadequate now [51, 52].

This study is one of the largest prospective cohort studies of COVID-19 mortality in Bangladesh covering 36% of total COVID-19 cases in the country during the study period. These results can help guide public health and clinical decision-making during future waves of the COVID-19 pandemic in Bangladesh and other low resource settings. They are particularly relevant for settings where risk factors for mortality may differ from those commonly cited in the literature from high-income countries, and the need for targeted interventions is more acute due to resource constraints. The results of the classification tree may be helpful for rapid clinical decision-making and provide a useful model for classifying high and low-risk patients at initial screening by first level health care providers. Finally, our results show that a physician's assessment of a patient's health status during the telehealth interview was highly predictive of mortality, demonstrating the potential value of the telehealth service. Harnessing the benefits of the telehealth system and optimizing care for those most at risk of mortality are key directions for future research.

## Supporting information

**S1 Fig. Directed acyclic graph showing hypothesized causal pathways of risk factors for COVID-19 mortality.**
(TIF)

**S2 Fig. Classification and regression tree for mortality among COVID-19 patients using age, gender, comorbidities, presenting symptoms, and physician's assessment of a patient's health status as predictors.** N represents the number of patients in the original data (as opposed to the resampled data used to create the tree). Blue indicates the predicted risk of mortality less than or equal to 2%; red indicates the predicted risk of mortality greater than 2% in the data.
(TIFF)

**S1 Table. Characteristics of the study population by availability of death outcome data.**
(DOCX)

## Acknowledgments

The authors want to thank the Ministry of Health and Family Welfare (MOHFW), Directorate General of Health Services (DGHS), Bangladesh, and the Institute of Epidemiology Disease Control and Research (IEDCR) for providing the data. We would also like to acknowledge the Aspire to Innovate (a2i) Programme, ICT Division, for implementing the telehealth program. We are extremely grateful to the telehealth doctors and Health Information Officers (HIO) without whom this study would not have been possible, as well as to the Bangladesh Association of Software and Information Services (BASIS), Synesis IT Ltd, Icon Information Systems Ltd. We also thank Shreya Roy for her assistance in preparation of the graphs in this paper.

## Author Contributions

**Conceptualization:** Ayesha Sania, Ayesha S. Mahmud, Anir Chowdhury, Shams el Arifeen.

**Data curation:** Daniel M. Alschuler, Tamanna Urmi, Shayan Chowdhury, Shabnam Mostari, Forhad Zahid Shaikh, Kawsar Hosain Sojib, Yiafee Khan.

**Formal analysis:** Daniel M. Alschuler, Seonjoo Lee, Tahmid Khan, Anir Chowdhury.

**Funding acquisition:** Ayesha Sania.

**Investigation:** Seonjoo Lee, Forhad Zahid Shaikh, Kawsar Hosain Sojib, Tahmid Khan, Yiafee Khan.

**Methodology:** Ayesha Sania, Daniel M. Alschuler.

**Resources:** Anir Chowdhury.

**Software:** Seonjoo Lee.

**Supervision:** Shams el Arifeen.

**Validation:** Tamanna Urmi, Shayan Chowdhury.

**Visualization:** Daniel M. Alschuler.

**Writing – original draft:** Ayesha Sania, Ayesha S. Mahmud, Tahmid Khan.

**Writing – review & editing:** Ayesha Sania, Ayesha S. Mahmud, Tahmid Khan, Shams el Arifeen.

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
