## [Decision Letter · Decision Letter 0]

19 Dec 2022

PGPH-D-22-01487

Risk factors for COVID-19 mortality among 290,488 telehealth patients in Bangladesh: a prospective cohort study

Dear Dr. Sania,

Thank you for submitting your manuscript to PLOS Global Public Health. After careful consideration, we feel that it has merit but does not fully meet PLOS Global Public Health’s publication criteria as it currently stands. Therefore, we invite you to submit a revised version of the manuscript that addresses the points raised during the review process.

Thank you for your submission, which we have assessed carefully. Below and attached you will find some suggestions from reviewers. In addition I have some suggestions myself:

- A major concern is the lack of ethical approval in Bangladesh. Please explain why this is not required; these are highly sensitive data(sets).

- 10% missing data on mortality outcomes is a very large proportion (much larger than the amount of deaths); the authors do mention this in the limitations. Seeing you will have most other data on these patients, have the authors done a comparison of the (baseline) characteristics to try to gauge the extent of the bias introduced by these missing data?

- The baseline table lacks information about physician assessment. Also, are there no data on hospitalisations? (And on what the physician's advice was, and whether this was followed up?)

- The added value of the regression trees is not clear to me. Please spell out the implications for the Bangladesh situation (or remove entirely)

- The methods lack clarity on merging of databases with death registry data

- The three different multivariable models based on the conceptual framework are difficult to understand (hence the question from reviewer 1, who clearly misunderstood that these were in fact multivariable models). I would suggest annotating the framework with the three models so it becomes more clear, and revise the text. 

- I support the concern from reviewer 1 about the large number of missing data and how they were treated. Did you consider multiple imputation?

- Please add rationale for the categorisation in periods. The data only go up to June 2021, why is this the case? Please also give some context about the covid vaccination status; was the entire cohort unvaccinated? 

- Did you consider including the time periods in one of your models? The time period effect on mortality in covid waves is very strong.

- The number of mobile phones in the discussion is difficult to interpret without a total population size to give indication of mobile phone penetration.

We look forward to receiving your revised manuscript.

Kind regards,

Sabine Hermans

Academic Editor

Journal Requirements:

a. Please clarify all sources of funding (financial or material support) for your study. List the grants (with grant number) or organizations (with url) that supported your study, including funding received from your institution. 

b. State the initials, alongside each funding source, of each author to receive each grant.

c. State what role the funders took in the study. If the funders had no role in your study, please state: “The funders had no role in study design, data collection and analysis, decision to publish, or preparation of the manuscript.”

d. If any authors received a salary from any of your funders, please state which authors and which funders.

2. In the online submission form, you indicated that your data will be submitted to a repository upon acceptance.  We strongly recommend all authors deposit their data before acceptance, as the process can be lengthy and hold up publication timelines. Please note that, though access restrictions are acceptable now, your entire data will need to be made freely accessible if your manuscript is accepted for publication. This policy applies to all data except where public deposition would breach compliance with the protocol approved by your research ethics board. If you are unable to adhere to our open data policy, please kindly revise your statement to explain your reasoning and we will seek the editor's input on an exemption. Please be assured that, once you have provided your new statement, the assessment of your exemption will not hold up the peer review process.

Additional Editor Comments (if provided):

Reviewers' comments:

Reviewer's Responses to Questions

**Comments to the Author**

1. Does this manuscript meet PLOS Global Public Health’s publication criteria? Is the manuscript technically sound, and do the data support the conclusions? The manuscript must describe methodologically and ethically rigorous research with conclusions that are appropriately drawn based on the data presented.

Reviewer #1: No

Reviewer #2: Yes

2. Has the statistical analysis been performed appropriately and rigorously?

Reviewer #1: No

Reviewer #2: Yes

3. Have the authors made all data underlying the findings in their manuscript fully available (please refer to the Data Availability Statement at the start of the manuscript PDF file)?

Reviewer #1: No

Reviewer #2: No

4. Is the manuscript presented in an intelligible fashion and written in standard English?

Reviewer #1: Yes

Reviewer #2: Yes

5. Review Comments to the Author

Reviewer #1: Overall, the manuscript is well-written. However, I have some comments available in the uploaded file.

Reviewer #2: This manuscript examined the contributing risk factors of mortality due to Covid-19. The study does provide some novel findings on identifying a few important risk factors of mortality due to Covid-19 among the Bangladeshi population.

However, there are a few issues which need to be improved before publishing this manuscript.

1. In the abstract the objective does not clearly state the problem for the research. Is this research only crucial because there is a lack of high-quality individual-level data?

2. In the main manuscript, though the introduction states the problem for the research in the second paragraph it explained the methodology of this research. The introduction should consist of the existing background information and the objective and rationale of the study, not the methodology.

3. The data is collected from the telehealth services which were self-reported by the patients. Did the researchers do any validity checks of the data to assess the accuracy of telemedicine assessments? If not, it should be listed as a limitation.

4. One of the most important findings of this research is physicians' assessment of the patients provides high predictability of mortality. But, how accurate this assessment is through telemedicine services? Its strength and limitations should be discussed in the discussion part.

5. Lastly, As both the incidence and mortality of Covid-19 is decreased and Bangladesh is successfully rolling out mass vaccination programmes, What is the implication of this study now? If there is any, then it should be discussed both in the introduction and discussion part.

6. PLOS authors have the option to publish the peer review history of their article (what does this mean?). If published, this will include your full peer review and any attached files.

**Do you want your identity to be public for this peer review?** For information about this choice, including consent withdrawal, please see our Privacy Policy.

Reviewer #1: No

Reviewer #2: No

---

## [Editor Report · Decision Letter 1]

4 May 2023

Risk factors for COVID-19 mortality among telehealth patients in Bangladesh: a prospective cohort study

PGPH-D-22-01487R1

Dear Dr. Sania,

We are pleased to inform you that your manuscript 'Risk factors for COVID-19 mortality among telehealth patients in Bangladesh: a prospective cohort study' has been provisionally accepted for publication in PLOS Global Public Health. Apologies for the delay in getting back to you. We appreciate the thorough revision based on our suggestions.

Best regards,

Sabine Hermans

Academic Editor